# Cultural transmission of attitudes and behaviours from parents, peers and grandparents

Monica Tamariz[1]*, Bradley Walker[2], Matthew Bennett[2], José Segovia-Martín[3], Nicolas Fay[2]

**1** Department of Psychology, Heriot-Watt University, Edinburgh, United Kingdom, **2** Department of Psychology, University of Western Australia, Perth, Australia, **3** School of Collective Intelligence, M6 Polytechnic University, Rabat, Morocco

\* M.Tamariz@hw.ac.uk

## Abstract

This study investigates how attitudes and behaviours are transmitted across generations and social networks, focusing on the relative influence of parents, grandparents, and peers. Building on the influential work of Cavalli-Sforza and Feldman (1982), we aimed to disentangle vertical and horizontal pathways of cultural transmission and assess their contribution to the stability and variation of cultural traits in a contemporary population. We conducted a large-scale survey involving 1905 university students in Australia and 4000 of their parents, grandparents, and friends. Participants reported their attitudes and behaviours across domains such as religiosity, politics, environmentalism, health, and leisure. Responses were analysed using factor analysis, path modelling, correlational analysis, and simulations based on additive transmission models. Our results show that cultural resemblance is strongest for religiosity, political orientation, environmentalism, and health behaviours. These traits exhibited clear vertical transmission from parents to children, with additional indirect influence from grandparents. Peer similarity was also evident, suggesting horizontal transmission and/or peer selection. Traits such as media use, music, and reading habits showed weaker familial resemblance and appeared more influenced by non-familial or contextual factors. Simulations confirmed that cultural traits are more likely to be adopted when shared by both parents and peers, though for some traits (especially left-wing political views and non-religiosity) external influences predominated. The findings demonstrate that cultural transmission is domain-specific and shaped by both family structure and social networks. Vertical and horizontal pathways contribute jointly, but their strength varies by trait. These results underscore the importance of integrating biological, psychological, and sociocultural factors to understand the persistence and evolution of beliefs and behaviours over generations.

**Data availability statement:** Data files are accessible from the OSF database. This is the URL: https://osf.io/v8hc7 and this is the DOI: https://doi.org/10.17605/OSF.IO/EJQ4U.

**Funding:** The author(s) received no specific funding for this work.

**Competing interests:** The authors have declared that no competing interests exist.

## Introduction

Attitudes and beliefs exert a profound influence on our behaviours. They shape our everyday actions as well as extreme behaviours that conflict with individual self-interest, such as vows of celibacy, suicide bombings, altruistic self-sacrifice, or rejection of effective medical treatments such as vaccines. The relationship between attitudes and beliefs on one hand and behaviours on the other is a strong, but complex one [1]. Beliefs –supposed understandings and non-trivial claims of facts and cause-effect relationships [2,3]–, and attitudes –evaluative judgements [4]– determine behaviour in complex ways that can also be guided by additional factors, such as social situational factors and personality [5].

A central aim of cultural evolutionary research is to explore the rich diversity of cultures within populations. It seeks to understand the origins of behaviours, beliefs, values, and attitudes, and how they are shared and transmitted through social networks. The specific cultural variants an individual holds—such as attending church, mosque, synagogue, or not, or identifying with left-wing, right-wing, or other political ideologies—are shaped by a variety of factors, including socio-cultural, biological, psychological, populational and environmental influences. Interpersonal social transmission plays a vital role in the dissemination of information within populations and is a foundational concept in cultural evolution [6,7]. A primary mechanism of this form of transmission is social learning—learning from others [8–10]. Cavalli-Sforza & Feldman [6] focused on three key pathways of cultural transmission: vertical (from parents to children), horizontal (among peers), and oblique (from unrelated adults, such as teachers).

### Cultural transmission pathways

**Vertical transmission.** Whereby cultural knowledge is passed from parents to children, is a powerful source of cultural influence, shaping traits such as political attitudes [11], religiosity [12,13], and racial attitudes [14]. Sustained parent-to-child transmission gives vertical transmission a unique role as a 'stabiliser' of culture, helping to preserve attitudes and beliefs within families and across generations [6]. Due to the formal similarities between the transmission of genetic and cultural material from parents to children, much of the research on vertical transmission has drawn on genetic models [6,7,13]. However, "reverse" vertical transmission, where children influence parents—such as daughters encouraging environmentally conscious consumption [15] or adolescents influencing their parents' attitudes towards immigrants [16]—can introduce novelty and disrupt family traditions, while still reinforcing parent-child resemblance. Vertical transmission tends to endure over time; parental influence persists and has a lasting impact throughout life [17–19].

**Horizontal transmission.** Or the transmission of knowledge between peers, is pervasive. This is evident for health-related behaviours such as diet, exercise, smoking, alcohol consumption, and environmental awareness [20–28]. However, horizontal transmission can be challenging to separate from homophily—the tendency to associate with similar others. Studies that account for homophily have found evidence supporting horizontal transmission as an independent process

[29,30], although the extent to which peer-to-peer attitudinal and behavioural similarities are due to transmission or homophily remains unclear.

**Oblique transmission.** Occurs when cultural knowledge is passed from older, non-related individuals—such as teachers or other social agents—to younger people. Cavalli-Sforza & Feldman [6] differentiate between two types of oblique transmission: "many-to-one," where the collective influence of a politically or religiously homogeneous community shapes younger individuals, often leading to cultural uniformity and lasting traditions; and "one-to-many," where a single influential figure, such as a teacher or celebrity, impacts many individuals. The latter can drive rapid cultural change, as successive figures influence younger generations. While this paper focuses primarily on vertical and horizontal transmission pathways, the effects of oblique transmission will be discussed in the concluding section.

## Additional explanations for shared cultural traits

Similarity in behaviour and attitudes among individuals is not only driven by social transmission; biological, psychological, and environmental factors can also contribute to shared cultural traits.

From a biological standpoint, genetics and epigenetics can help explain why the attitudes and behaviours of related individuals, such as parents and children, tend to be more similar than those of unrelated individuals. Twin, adoption, and family studies have been instrumental in disentangling the contributions of genetics to these similarities [31]. Dual-Inheritance Theory also posits that cultural and genetic factors shape an individual's attitudes, beliefs, and behaviours [32]. However, an additional challenge in separating these influences, particularly in parent-child transmission, is gene-culture coevolution, which suggests that dynamic interactions between genetic and cultural factors guide individuals' attitudes, beliefs and behaviours [33–36]. Moreover, the relative influence of biological and cultural factors can vary depending on the specific cultural traits in question [33].

In psychology, homophily—the tendency to befriend or partner with individuals who are similar to us—promotes the alignment of attitudes and behaviours among friends, spouses, and other social groups [37–39]. In evolutionary biology, this phenomenon is known as assortative mating for couples, and in other fields, it is referred to as peer selection, for friends and peers. This process is contrasted with cultural transmission, in which individuals *influence* another's behaviours.

Finally, environmental factors such as shared living conditions, exposure to the same media, and even to similar climates [40,41] can independently contribute to similarities in attitudes and behaviours. Even in the absence of cultural transmission, individuals may be more likely to befriend or to partner with individuals they meet in their common environment, or to develop similar traits due to shared conditions, emphasizing, once more, that behavioural similarity does not always imply cultural transmission.

The complex interplay between these social, biological, psychological, and environmental factors makes it clear that behavioural resemblance is the product of multiple, interacting influences. For example, a child might inherit a behaviour or attitude (biologically and/or culturally) from her parents, and then select friends who resemble herself and her parents (psychologically, influenced by homophily). As such, resemblance should not be automatically attributed to a single cause, such as cultural transmission alone, but rather understood through a multifactorial lens.

## Cavalli-Sforza et al.'s seminal study

Cavalli-Sforza and Feldman's [6] genetic model of cultural transmission is among the foundational works in cultural studies, helping explain cultural trait homogeneity as well as the maintenance of diversity in populations. Using a correlational study, Cavalli-Sforza and colleagues [13] explored the relative cultural influence of parents and peers – they quantified the similarity between undergraduate biology students and their parents and peers in the cultural domains of religion, politics, entertainment, sports, beliefs, and habits. They asked students to complete a survey that included questions such as whether they attend church or are a registered voter and instructed the students to recruit their parents and two friends

(one male and one female) to complete the same survey. Vertical transmission was estimated based on the data from students who returned responses from both parents ($N=203$), and horizontal transmission was estimated based on the data from students who returned responses from two friends ($N=98$).

Cavalli-Sforza and colleagues [13] found statistically significant correlations for specific questions within each of the tested domains, both for vertical and horizontal transmission. Vertical correlations were the highest and were particularly high for religion ($r=.57$) and politics ($r=.32$); the rank order of vertical correlations from highest to lowest was: religion > politics > entertainment > sports > beliefs > habits. Horizontal correlations tended to be lower, ranging from $r=.05$ to $r=.20$. However, the horizontal correlations were slightly higher than the vertical correlations for the domains of sports and beliefs. The rank order of horizontal correlations from highest to lowest was: religion > sports/politics > entertainment/ beliefs > habits. As their results were based on correlational analyses, some of the covariance found may have been shared between the vertical and horizontal correlations, meaning that some portion of the observed 'influence' was not uniquely vertical or horizontal. The present study will examine transmission in similar, updated cultural domains and in expanded social network using different analytic approaches to examine the unique effects of vertical and horizontal transmission on attitudes, beliefs and behaviours.

### The role of grandparents

Grandparents represent a potential source of vertical transmission, encompassing, as in the case of parents, genetic and cultural elements. Grandparents often transmit knowledge and values directly to their grandchildren, sharing family wisdom, offering encouragement, and shaping goals [42–44]. However, in Western societies, where the nuclear family structure predominates and often excludes the grandparental generation from daily household interactions, the direct influence of grandparents may be more limited [45]. In such cases, grandparents may still exert influence indirectly, through their impact on the parents. Grandparent influence as an additional component of vertical transmission may slow cultural change [6]. Alongside parents and friends, this study will explore the role of grandparents, a second source of vertical transmission, in shaping the attitudes and beliefs of a student population.

### The present study

Following from the research by Cavalli-Sforza and Feldman [6,13], this study addresses, 40 years on, similar questions to those they posed [13]:

• What is the relative importance of vertical and horizontal cultural pathways of transmission?

• How does transmission operate across a variety of cultural attitudes and behaviours, from religion and politics to reading habits and musical preferences?

• Which mechanisms are responsible for the stability of cultural phenomena over time?

Additionally, this study builds on Cavalli-Sforza et al.'s (1982) work in three key ways. First, it isolates the unique agreement of students' attitudes and behaviours with their parents, as compared to the agreement with their peers. Second, it quantifies the relative strength of potential vertical, horizontal and other influences such as additional social connections, media, or personal discovery and invention on an individual's attitudes and behaviours. Third, the study incorporates grandparents, providing insights into a broader social network and allowing for the examination of direct and indirect vertical transmission across three generations. Finally, our sample of respondents was more than six times greater than Cavalli-Sforza and colleagues' [13]. Our additional research questions are:

• What are the unique and joint contributions of vertical and horizontal pathways on attitudes and behaviours?

• How much of the variance in attitudes and behaviour can be attributed to peer and parent influence, and how much to third factors?

- What influence do grandparents exert on grandchildren's attitudes and behaviours?

- To what extent can we isolate the effects of cultural transmission on shared cultural traits from other explanations of agreement or similarity?

This survey-based study explores a new, modern data set from multiple perspectives to capture transmission and other causes of similarity and influence between a sample of students and their parents, friends and grandparents.

## Methods

### Participants

A total of 6147 participants completed our questionnaire. This included 1905 students from the University of Western Australia (UWA), in Perth, Australia. All students received study credit for their participation. The students' ages ranged from 16 to 87 ($M = 21.31$, $SD = 6.32$), and genders were 72.2% women, 27.1% men, and 0.7% other. An additional 4242 people were recruited by the participating students. These participants were family and friends of the student participants. Of these, 4000 were parents, grandparents or friends of the participating students (Table 1). The remaining 242 participants were other relatives whose data were not used. The retained participants constituted 2492 networks (groups including at least a single student, and optionally their family and their friends). 722 networks only comprised the student with no family members or friends and were therefore excluded (for network composition see Table A in S1Text). After exclusions, data from 5183 respondents (from 1285 networks) were analysed. There were some missing responses ($N = 3305$, 2.0% of the total including all participants and both questionnaires from students who completed the questionnaire twice; see Procedure), which were imputed using expectation maximization.

Our reliance on a student convenience sample limits the generalizability of the findings, as students may differ from the broader population in demographics, network structure, and social experiences. Data from a single Australian university also reflects limited geographical and cultural diversity, which may constrain the applicability of the results to other settings. In addition, there was a notable gender imbalance, with women being overrepresented, which may influence the observed patterns of social transmission. Recruitment of family and friends by student participants was relatively low, with many students not recruiting anyone and, among those who did, most recruiting only a small number of contacts. Finally, excluding networks where no family or friends participated may have introduced attrition bias, potentially underrepresenting socially isolated individuals and networks with different transmission dynamics.

### Materials

We developed a survey (S2 Text) inspired by Cavalli-Sforza and colleagues [13], but updated to align with contemporary developments and avoid outdated content. Their survey included 38 questions about religion, sports, politics,

**Table 1. Participant descriptives. Participant number, gender and age descriptive statistics.**

| Relation | N | Women | Men | Other | Age (mean) | Age (SD) |
|---|---|---|---|---|---|---|
| Student | 1905 | 71.8% | 27.7% | 0.6% | 21.30 | 6.15 |
| Friend | 1822 | 60.9% | 38.6% | 0.6% | 21.60 | 7.42 |
| Mother | 835 | 99.5% | 0.5% | 0.0% | 51.72 | 6.83 |
| Father | 696 | 0.4% | 99.6% | 0.0% | 53.89 | 7.18 |
| Maternal Grandmother | 260 | 99.2% | 0.8% | 0.0% | 74.77 | 9.26 |
| Maternal Grandfather | 161 | 1.9% | 97.5% | 0.6% | 76.72 | 8.26 |
| Paternal Grandmother | 130 | 95.2% | 4.8% | 0.0% | 74.94 | 9.71 |
| Paternal Grandfather | 96 | 1.1% | 98.9% | 0.0% | 78.40 | 8.94 |
| All | 5905 | 62.6% | 37.0% | 0.03% | 35.38 | 20.64 |

entertainment, habits and beliefs (S3 Text); about half of them asked about observable behaviours (e.g., Do you attend church? Do you attend the movies?), and the other half about attitudes or beliefs (e.g., Are you a Catholic? Do you believe in UFOs?). Most questions had binary Yes/ No answers.

Our survey (S2 Text) included 27 questions about religion, sports, politics, environmentalism, health, entertainment, and socialisation. Within each topic, we included questions about attitudes (e.g., Do you believe in a god or higher power? How much do you enjoy listening to music?) and behaviours (e.g., How many hours do you spend listening to music? Do you take action to help the environment in any of the following ways?). We made some updates, e.g., removing questions about horoscopes, UFOs and ESP and adding questions about social media use and different answer types to capture more nuanced responses. Twenty of the questions had Likert-scale answers. Seven (Q2, Q5, Q7, Q8, Q11, Q15 and Q27) required a text answer. In questions Q2, Q5, Q7, Q8, and Q15, respondents could select a single option from a list or type their answer in a text box. In questions Q11 and Q27, respondents could select as many options as they wished from a list.

## Procedure

The study received approval from the University of Western Australia Ethics Committee. Participants viewed an information sheet, gave consent to take part by clicking on "agree" in the survey after the consent statements, and were debriefed afterwards. All methods were performed in accordance with the guidelines from the National Health and Medical Research Council/Australian Research Council/University Australia's National Statement on Ethical Conduct in Human Research. Data was collected between 30 March 2020 and 26 May 2023.

This study was conducted online. Participants completed the survey and sent a link to their parents, grandparents and two close friends asking them to complete the same survey online. Student participants completed the survey again a few weeks later to allow us to check the reliability of their answers.

## Analyses

**Factor analysis.** Rather than analyse every item in the questionnaire separately, we first reduced the data via an exploratory factor analysis (maximum likelihood estimation with Promax rotation), using the psych package [46] in R [47]. All numeric items were included in the factor analysis, including non-numeric items that were amenable to re-coding as numeric (see S4 Text). Examination of the scree plot and parallel analysis suggested the extraction of eight factors. For subsequent analyses, we used the estimated factor scores for these eight factors, rather than the raw data (for descriptives by question see S5 Text).

**Path analysis.** To test the unique contributions of vertical and horizontal sources of influence, we used path analysis. This allowed us to model the vertical and horizontal influences on students simultaneously, while controlling for each other. As many networks within our data did not include grandparents, we first modelled the influence of parents and friends on students (i.e., two direct effects). To be included in this first set of path analyses, networks needed to include at least one parent and at least one friend, in addition to the student; 582 such networks were analysed. Second, for 223 of those networks which additionally included at least one grandparent, we adapted the model to include grandparents. Grandparents were allowed to influence students directly, as well as indirectly via parents (i.e., a mediation relationship). For networks that had multiple parents, friends or grandparents, we averaged the scores within each relationship; this simplified the models and allowed us to include more networks than would have been possible with relationships separated out (e.g., requiring both parents and all four grandparents). As people with the same relationship in the same network (e.g., the two parents) tended to have correlated responses (see section 3.3 'Correlations and Resemblance'), we felt that this was justified. Path analyses were conducted separately for each factor using the lavaan package [48] in R [47]. Statistical significance was determined via bootstrapping (10k resamples).

**Correlation analysis and resemblance.** To examine the strength of the associations or similarity between the responses given by different social agents (e.g., students and mothers), we conducted, like [49], Pearson's correlations.

Given the high multicollinearity and the non-normality of response distributions (see histograms in S5 Text), statistical significance was estimated using a Monte Carlo method. For each factor, and for each pair of relationships (e.g., students and mothers), we obtained the z-score of the veridical correlation coefficient (e.g., between the factor weights of students and the factor weights of their mothers) in a distribution of 10,000 scrambled correlations (e.g., between the factor weights of students and the factor weights of random mothers from our sample).

We call this z-score 'Resemblance'. This measure can be related to the concept of broad heritability in biology –the degree to which offspring resemble their parent in a particular phenotypic trait (such as intelligence or height) more than they resemble a random member of the population. Resemblance values reflect whether, e.g., students' responses are more similar to their mothers' responses than to the responses of a random mother from our sample. Like biological heritability, Resemblance depends on the variance in a population. For example, if everybody in a population shares the same religion, there is similarity across the board, but students resemble their mothers as much as a random mother in the population, and Resemblance is very low.

This Resemblance measure is agnostic as to the mechanisms and provide an exploratory picture that allows us to examine: similarity between the responses of age-peers (via horizontal pathways), e.g., the student and their friends; similarity between the responses of agents who have selected each other's company: assortative mating, e.g., mother and father, paternal grandmother and grandfather, or peer-selection, e.g., the student and their friends; and similarity between the responses of agents who do not have a direct connection, such as the student's friends and the student's grandparents.

These patterns of similarity allow us to formulate hypotheses not only about possible influences on students, but also about the distribution of attitudes and beliefs in society. For example, high correlations among all agent pairs, including those with no direct connections, suggests a clustered social network.

**Additive transmission of vertical and horizontal transmission and bias estimation.** To estimate the relative contribution of vertical, horizontal and other sources to the student responses, we took inspiration from Cavalli-Sforza and colleagues' [49] additive model of transmission and augmented it with a simulation. While they constructed parental types using biological parents (mother and father), we considered two cultural parental types: vertical (mother and father) and horizontal (two friends). Using only networks that had answers from the student and at least one parent and one friend, the mother and father factor weights were averaged into a Vertical component, and the two friend factor weights into a Horizontal component. Then the student, Horizontal and Vertical factor weights were dichotomised: if greater than or equal to 0 (0 being the mean of the factor weights), they were assigned value 1 and assumed to have the trait at hand, e.g., high religiosity, right-wing political orientation, etc; if less than 0, they were assigned value 0 and assumed to not have the trait (or to have the opposite trait: low religiosity, left-wing political orientation, etc.). A limitation of this method is the fact that the dichotomization of continuous factor scores, while faithful to the original method [49], involves a loss of information.

Networks were divided into parental types: VH (capital V and H: Vertical and Horizontal values have the trait), Vh (capital V only: only Vertical has the trait), vH (capital H only: only Horizontal has the trait) and vh (lowercase v and h: neither has the trait). We counted the number of networks with each parental type ($P_i$) and obtained the transmission coefficients $B_i$ by dividing the number of students with the trait in each parental type ($N_i$) by $P_i$. The coefficients represent the proportion of students who have the trait within each parental type (Table 2).

We define three biases: Vertical bias captures the tendency for students to resemble their parents in the trait at hand; Horizontal bias, for students to resemble their friends; and Other bias, for students to resemble other sources not captured in the data, such as other people (i.e., teachers, other relatives), media, individual invention, etc. (Table 2). For instance, in parental type Vh, parents have the trait but friends do not; students with this parental type learn the trait from their parents or from other sources, but not from friends, therefore only Vertical and Other biases apply.

A maximum likelihood estimation simulation was used for each factor. The simulation was based on four equations:

$$N_{VH} = P_{VH} * (Other + Vertical + Horizontal)$$

**Table 2. Cultural parental types and relevant biases.**

| Cultural Parental Type | $P_i$ | $N_i$ | $B_i$ | Relevant Biases |
|---|---|---|---|---|
| VH – Vertical & Horizontal | $P_{VH}$ | $N_{VH}$ | $B_{VH} = N_{VH} / P_{VH}$ | Other + Vertical + Horizontal |
| Vh – Vertical only | $P_{Vh}$ | $N_{Vh}$ | $B_{Vh} = N_{Vh} / P_{Vh}$ | Other + Vertical |
| vH - Friends only | $P_{vH}$ | $N_{vH}$ | $B_{vH} = N_{vH} / P_{vH}$ | Other + Horizontal |
| vh – Neither | $P_{vh}$ | $N_{vh}$ | $B_{vh} = N_{vh} / P_{vh}$ | Other |

$$N_{Vh} = P_{Vh} * (Other + Vertical)$$
$$N_{vH} = P_{vH} * (Other + Horizontal)$$
$$N_{vh} = P_{vh} * (Other)$$

We inserted the veridical $N$ and $P$ values calculated with our data in the equations and simulated over values for the Other, Vertical and Horizontal biases to find the parameter value combination that returned the lowest amount of error for each factor.

## Results

### Descriptive statistics

The summary of responses to each question are in Fig 1. Response distributions varied by question (but within each question, all social agents showed broadly similar distributions, see S5 Text). Our sample showed diversity in religious beliefs, a slight left-leaning skew in political orientation and high environmentalism, but lower levels of active engagement with religion, politics and environmentalism; and mostly positive attitudes towards healthy habits, with little smoking and alcohol use and healthy diets.

The correlations between the responses by each pair of social agents are in Fig 2. The highest correlation is between the responses of students at time = 1 and time = 2, which indicates high reliability. Correlations between the maternal, and then the paternal grandparents are also among the highest. Students (time = 1) are most correlated with their friends and their mother. (For correlations by question see S5 text.)

### Factor analysis

An exploratory factor analysis was used to identify the underlying dimensions that explain the pattern of correlations to the survey questions, and to simplify the interrelated survey questions into a smaller number of coherent factors. The factor analysis yielded an eight-factor solution that demonstrated an acceptable overall fit to the data, with good values for the CFI (.95) and RMSEA (.04), and a TLI (.89) that was slightly below the conventional cutoff of.90. The factor loadings and communalities ($h^2$) are presented in S6 Table. We named the factors based on the most salient loadings: Religiosity (with high scorers more likely to believe in a god or higher power, identify with a religion, and attend religious ceremonies), Politics (with high scorers more likely to identify as right-wing and prefer right-wing political parties), Environmentalism (with high scorers more likely to be concerned for the environment, have a high number of pro-environmental habits, and be politically active), Health (with high scorers exercising more frequently and placing more importance on a healthy lifestyle), Music (with high scorers spending more time and gaining more enjoyment from listening to music), Reading (with high scorers spending more time and gaining more enjoyment from reading), Screen (with high scorers spending more time and gaining more enjoyment from video media, and spending more time on social media), and Social (with high scorers having more close friends and engaging more with social activities).

Table 3 shows the Vertical (averaged mother-student and father-student) and Horizontal (averaged friend1-student and friend2-student) correlations of factor loadings. Overall, Horizontal correlation is higher than Vertical. By factor, Vertical correlations are stronger for Religiosity and Politics, while Horizontal correlations are stronger for Screen, Social, Music and Reading.

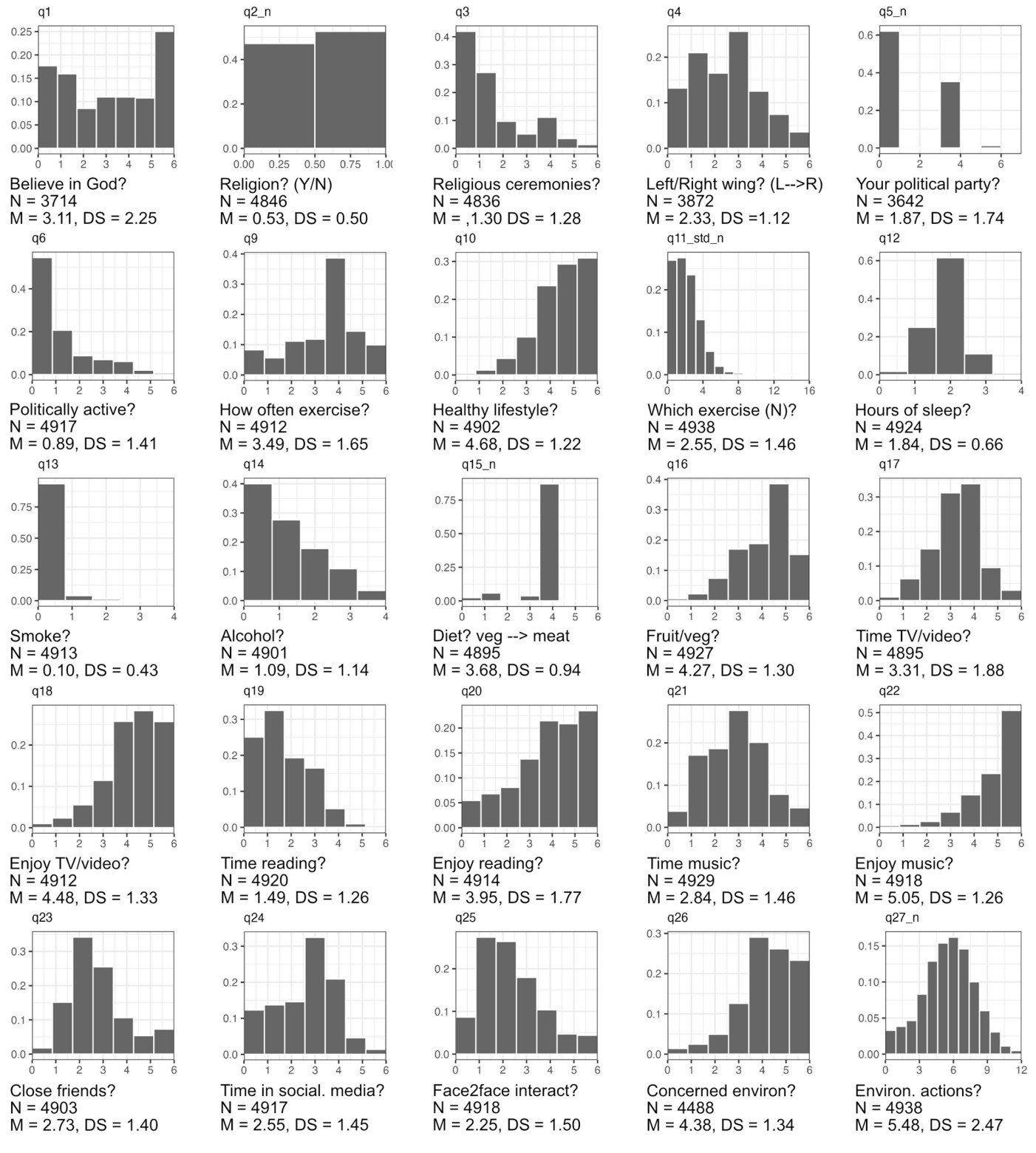

**Fig 1. Descriptives of the responses to each question aggregated over all respondents.**

|  | student (time=1) | student (time=2) | paternal grandM | paternal grandF | mother | maternal grandM | maternal grandF | friend2 | friend1 |
|---|---|---|---|---|---|---|---|---|---|
| father | .579*** | .592*** | .588*** | .564*** | .663*** | .529*** | .537*** | .518*** | .521*** |
| friend1 | .650*** | .660*** | .448*** | .472*** | .548*** | .463*** | .427*** | .601*** |  |
| friend2 | .644*** | .655*** | .417*** | .428*** | .556*** | .443*** | .407*** |  |  |
| mat. grandF | .477*** | .489*** | .583*** | .648*** | .583*** | .716*** |  |  |  |
| mat. grandM | .514*** | .559*** | .585*** | .540*** | .617*** |  |  |  |  |
| mother | .621*** | .634*** | .552*** | .523*** |  |  |  |  |  |
| pat. grandF | .502*** | .620*** | .700*** |  |  |  |  |  |  |
| pat. grandM | .485*** | .529*** |  |  |  |  |  |  |  |
| student(time=2) | .905*** |  |  |  |  |  |  |  |  |

**Fig 2. Pearson's correlation coefficients of responses to all questions by each pair of relations.** Higher values have darker shades. All $p < 10^{-31}$.

**Table 3. Correlational estimates of vertical and horizontal factor loadings.**

| Factor | Vertical | Horizontal |
|---|---|---|
| Religiosity | 0.59*** | 0.41*** |
| Politics | 0.31*** | 0.24*** |
| Environ. | 0.30*** | 0.29*** |
| Health | 0.23*** | 0.22*** |
| Screen | 0.17*** | 0.23*** |
| Social | 0.14† | 0.23 † |
| Music | 0.10† | 0.12** |
| Reading | 0.08 | 0.11 |
| *Overall* | *0.14*** | *0.27*** |

† p<.10. * p<.05. ** p<.01. *** p<.001.

Our correlation-by-factor ranking (Table 3) was generally similar to that in [49] (Table 4). In both studies, the strongest correlations among agents were found for Religion, followed by Politics, with Entertainment being the least correlated of the factors that are comparable across studies. This consistency in results between the US in the 1980s and Australia in the 2020s together with the religious diversity in our samples, especially among Australian responses, comprising 14 distinct religious beliefs (Table A in S7 Text) suggests that religion and politics are culturally stable, persistent traits and strongly shared both horizontally and vertically in social networks, at least in these two Western countries.

It is worth noting the higher correlations found for Religiosity and Politics between students and their friends in our sample than in [49] (Table 5). The reason for this is unclear. It may be due to shifts in communication technology over the

**Table 4. Correlation rankings obtained by the current study and [49].**

| |
|---|
| Current study (8 factors): |
| Vertical: Religiosity > Politics = Environmentalism > Health > Screen > Social > Music > Reading |
| Horizontal: Religiosity > Environmentalism > Politics > Screen > Social > Health > Music > Reading |
| Cavalli-Sforza et al. (1982) (6 topics): |
| Vertical: Religion > Politics > Entertainment > Sports > Beliefs > Habits. |
| Horizontal: Religion > Sports > Politics > Entertainment = Beliefs > Habits |

**Table 5. Correlation values for the two most correlated factors in our study and in [49].**

| | Our results | | Cavalli-Sforza et al. (1982) | |
|---|---|---|---|---|
| | Parent-student | Friend-student | Parent-student | Friend-student |
| Religiosity | 0.57 | 0.20 | 0.59 | 0.41 |
| Politics | 0.32 | 0.16 | 0.31 | 0.24 |

past 40 years, and this may have increased the degree to which friends influence each other: Communication frequency appears to moderate transmission [50,51], and more opportunities to communicate may lead to greater influence. Another explanation may lie in a shift toward less authoritarian parenting styles [52], which allows more freedom for children [53], increasing the opportunity for outside influences. However, if parenting style shift is responsible for the change, it is unclear why vertical transmission has not been reduced alongside the increased horizontal transmission. Peer selection could be a factor here, e.g., with international students in the Australian sample recruiting same-nationality friends who (due to nationality) share their religious background.

## Path analysis

Path analysis was used to test the theoretically informed model of vertical and horizontal influence on beliefs and attitudes. This approach allowed us to quantify both the direct and indirect effects of these different sources of influence. Considering all the networks that included grandparents, we ran two different path analysis models (Fig 3), Model 1 with students' scores predicted just by parents and friends (582 students, 430 mothers, 175 fathers, 877 friends), and Model 2 with students' scores additionally predicted by grandparents, both directly and indirectly via parents (223 students, 187 mothers, 175 fathers, 364 friends, 77 paternal grandmothers, 148 maternal grandmothers, 66 paternal grandfathers, 99 maternal grandfathers). The different factors were analysed separately. Model 1 follows [49], while Model 2 additionally includes grandparents. Correlations for each factor are in Table 6 and standardised coefficients in Table 7.

The path analyses indicate that most factors are subject to influence: Religiosity, Politics, Environmentalism, Health, Screen and Social all show relationships between students' scores and the scores of their parents, grandparents and friends. Only Music and Reading show little evidence for social influence. The sequence of factors ordered by the strength of the coefficients is similar across all relationships, suggesting that Religiosity, Politics, Environmentalism and Health are more subject to social influence, both vertical and horizontal, than Music, Reading, Screen and Social.

Models 1 and 2 both show evidence for vertical and horizontal influence. For most factors, the influence of parents is similar to the influence of friends. However, parents' influence is notably stronger for Religiosity, and friends' influence is notably stronger for Screen and Social. These patterns make sense given that parents frequently induct their children into their religions [12,54–56], and engagement with screen media and social behaviours often involve friends more than parents [57].

Path analysis also finds evidence of vertical influence from grandparents, both on parents and on students. Adding the grandparents to our model improved the explanatory power for all factors. The fact that grandparents predicted parents' scores on all factors but Reading (typically with similar strength to the parent-student relationship) indicates that vertical

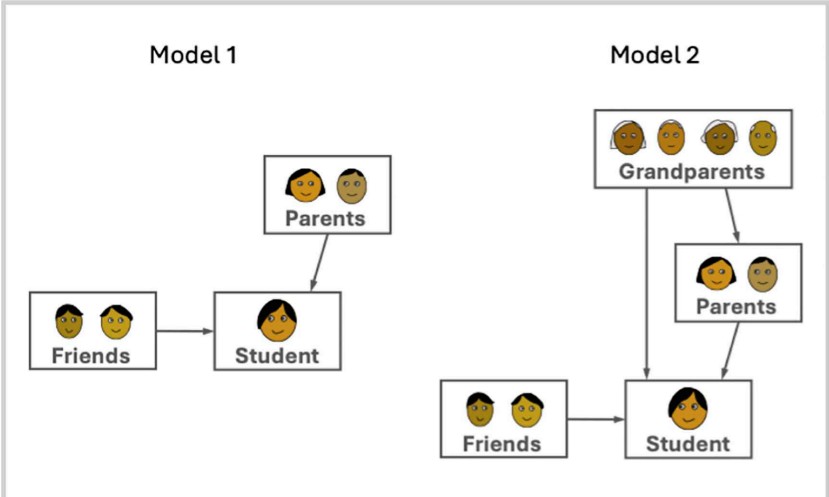

**Fig 3. The two Models used in the path analysis.**

**Table 6. Pearson correlations between students, parents, friends and grandparents for the 8 factors.**

| Relationship | Religiosity | Politics | Environ. | Health | Music | Reading | Screen | Social |
|---|---|---|---|---|---|---|---|---|
| **Model 1** (582 networks) | | | | | | | | |
| Parent–Student | .61*** | .35*** | .31*** | .26*** | .07† | .07 | .17*** | .19*** |
| Friend–Student | .45*** | .34*** | .35*** | .30*** | .10* | .07 | .24*** | .26*** |
| **Model 2** (223 networks) | | | | | | | | |
| Parent–Student | .63*** | .40*** | .47*** | .39*** | .15* | .16* | .25*** | .19** |
| Friend–Student | .49*** | .38*** | .45*** | .24*** | .08 | .04 | .27*** | .25*** |
| Grandparent–Parent | .54*** | .29*** | .46*** | .34*** | .23*** | .11† | .21** | .21** |
| Grandparent–Student | .43*** | .19** | .39*** | .29*** | .12† | .06 | .11† | .08 |

† p<.10. * p<.05. ** p<.01. *** p<.001.

influence is ongoing and/or long-lasting. There are direct relationships between grandparents' and students' scores for Environmentalism and Health, but the larger influence of grandparents is indirect, via parents, with indirect relationships for Religiosity, Environmentalism, Health, Politics and Screen. This indicates that grandparents have a non-trivial influence on the student's attitudes and behaviours, mediated by the parents; the smaller influence compared to parents may be explained by lower levels of contact across two generations than across one generation in nuclear families [45,58].

## Resemblance

While path analysis focuses on similarity between vertically and horizontally connected agents, a more thorough analysis of similarity and, potentially, transmission among all agent pairs provides further insights into the behaviour of cultural traits in the social network. To quantify the degree of similarity between all relation pairs, Resemblance (the z-score of each pairwise correlation calculated using a Monte Carlo approach) was obtained for each factor, for every pair of relationships (Fig 4). Results reveal diversity across factors and social agents. The strongest ties are found for Religiosity,

**Table 7. Standardised coefficients and $R^2$ for each path analysis model.**

| Factor | Model | P→S | F→S | G→S (total) | G→S (direct) | G→P | G→P→S (indirect) | $R^2$ |
|---|---|---|---|---|---|---|---|---|
| Religiosity | 1 | .52*** | .27*** | | | | | .44 |
| | 2 | .50*** | .29*** | .33*** | .06 | .54*** | .27*** | .45 |
| Politics | 1 | .31*** | .30*** | | | | | .21 |
| | 2 | .32*** | .31*** | .16* | .06 | .29*** | .09** | .23 |
| Environment. | 1 | .23*** | .28*** | | | | | .17 |
| | 2 | .28*** | .29*** | .30*** | .17** | .46*** | .13*** | .30 |
| Health | 1 | .22*** | .26*** | | | | | .14 |
| | 2 | .31*** | .10 | .26*** | .15* | .34*** | .11** | .18 |
| Music | 1 | .06 | .09 | | | | | .01 |
| | 2 | .13 | .05 | .11 | .08 | .23** | .03 | .03 |
| Reading | 1 | .06 | .06 | | | | | .01 |
| | 2 | .15* | .02 | .06 | .05 | .11 | .02 | .03 |
| Screen | 1 | .15*** | .22*** | | | | | .08 |
| | 2 | .20** | .23** | .10 | .06 | .21*** | .04* | .10 |
| Social | 1 | .16*** | .25*** | | | | | .10 |
| | 2 | .15* | .22*** | .04 | .01 | .21** | .03 | .08 |

Note. S = Student, P = Parents, F = Friends, G = Grandparents. Model 1 includes only students, parents and friends; Model 2 additionally includes grandparents (see Fig 3). $R^2$ shows the variance explained in students' scores by the model.

* $p < .05$. ** $p < .01$. *** $p < .001$.

Environmentalism and Politics. There are many vertical links in Religiosity, Environmentalism, Politics and Health, but not in the other four factors. As for horizontal links, we find significant Resemblance between mother and father, and between student and at least one friend for most factors. Grey lines connecting agents who have no obvious social link, such as those joining a friend with the student's parents or grandparents, are common only for Religiosity, Environmentalism and Health. (For Resemblance values by question, see Figure A in S8 Text)

Exploratory Resemblance results (Fig 4) corroborate the picture from the path analysis. In addition, they show consistent horizontal associations between mother and father across all factors, and between each grandparent couple and the student and the two friends for all factors except reading. Unlike vertical links, which may be at least partially due to genetic inheritance, these horizontal associations must be explained by social mechanisms such as cultural transmission (mutual or directional), similarity-based homophily, or shared experience.

Fig 4 also reveals strong Resemblance between all pairs of agents in the social network, including those that have no direct connection such as the student's friends with their parents and grandparents, for Religiosity and Politics —as well as Environmentalism and Health, which are not clearly covered by [49]. This suggests that our Australian population clusters along those dimensions. For example, responses to text question 2 "What is your religion?" (S7 Text) yielded very strong Resemblance among all agents, including, e.g., the student's friends with the students' parents and grandparents. This indicates the presence of religious clusters in society, e.g., Protestant students tend to have Protestant family and friends, Jewish students, Jewish family and friends, etc. We suggest mechanisms behind this pattern in the Discussion.

**Additive transmission and partitioning the sources of contagion: Vertical, Horizontal and Other biases**

To estimate how much of the variance in attitudes and behaviour can be attributed to peer and parent influence, and how much to third sources not covered in our survey, we followed Cavalli-Sforza and colleagues' genetic-inspired analysis [49]. First we ran an additive model to establish the prevalence of cultural traits among students depending on whether

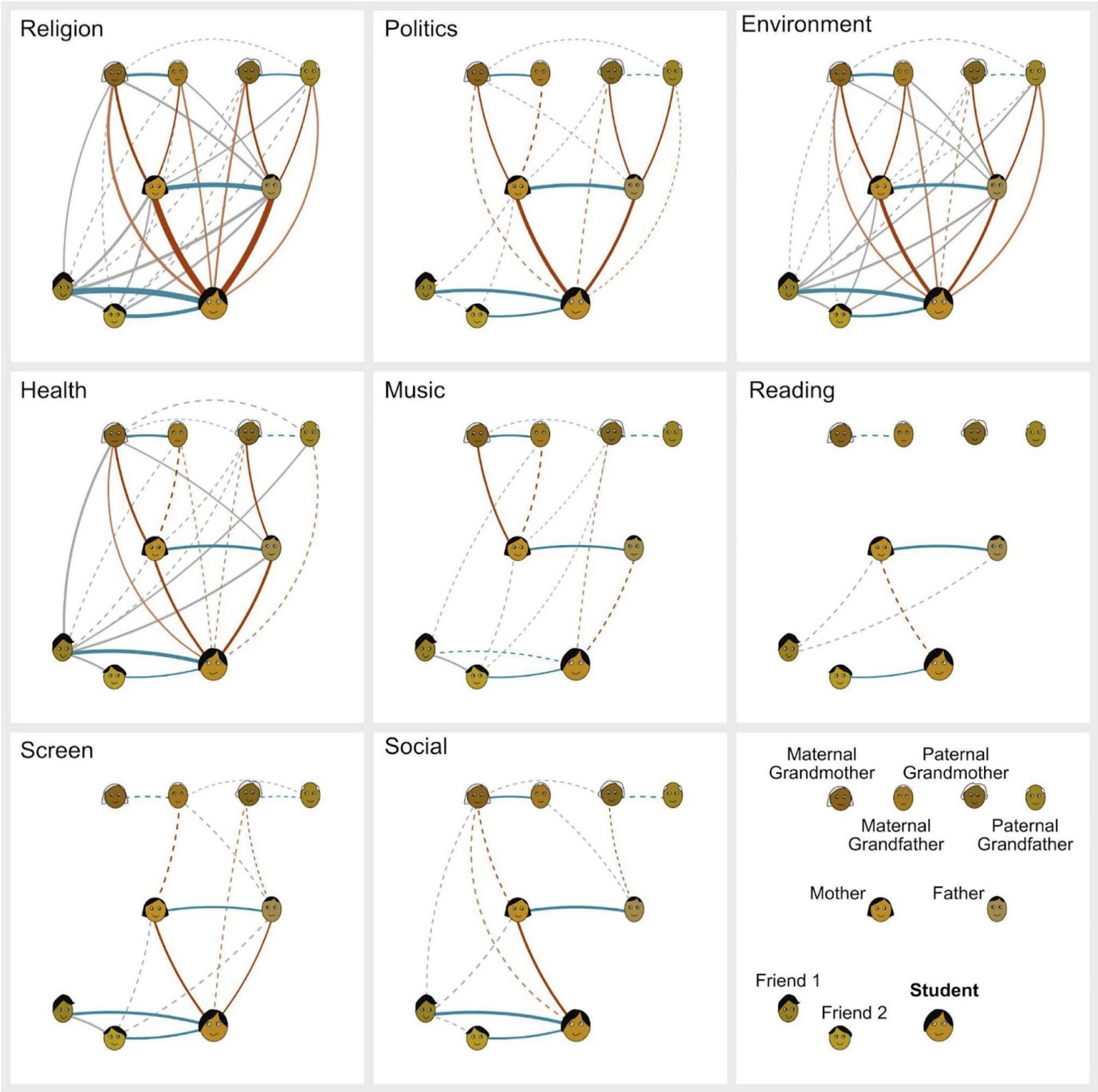

**Fig 4. Resemblance between each pair of social agents for the 8 factors.** (Bottom left panel identifies agents). Line width is proportional to z score. Solid lines: z > 3.3, p < 0.001.; dashed lines: z > 1.96, p < 0.05. Red lines: direct vertical (parent-student and grandparent-parent). Orange lines: indirect vertical (grandparent – student). Blue lines: horizontal (friend-student, couples). Grey lines: other connections.

their parents, their friends or both had the trait. Only the 582 networks in which the student had at least one parent and one friend in the data were included ($N=2090$). We dichotomised the scores (see 2.4.4) for each factor and conducted the following analyses for the trait and the opposite trait within each factor (Table 8). (See Figures A and B in S10 Text for equivalent results for each question.)

Next, we obtained $P_i$ values, the number of networks of each parental type and, from them, the transmission coefficients $B_i$, the proportion of students with the trait or opposite trait in each parental type (see an example in Table 9). Fig 5 shows $B_i$ values for networks in which the student has the trait. The $B_i$ values for the opposite trait would be complementary to these: the values in the figure subtracted from 1 and assigned to the inverse parental type (all values are in Table A in S9 Text and Figure A in S9 Text).

For instance, trait Environmentalism (Table 9) shows additivity: when more relations have the trait, the students are more likely to have it too (higher B). If the student's parents and peers both have the trait (Parental type VH), the transmission coefficient $B=.712$ is higher than if only the parents (Vh, $B=.513$) or only the peers (vH, $B=.541$) have the trait. And if neither parents nor peers (vh) have the trait, B is lowest ($B=.384$).

Fig 5 reveals differences in transmission coefficients across factors. For example, a large majority of students enjoy music (and, conversely, very few students have the opposite trait "not enjoy music"), and this is the case across parental types. In Politics (right-wing trait), a minority of students have right-wing political views (and, conversely, many have left-wing views), but here we also observe an additive effect of parental type: if *both* their parents and friends are right-wing, students are more likely to be right-wing than if *only* their parents or *only* their friends are right-wing. Most factors present at least some additive effects.

Looking at the relative influence of parents and friends (Fig 5), overlaps in most bootstrapped 95% confidence intervals indicate no difference, except for Religiosity. For this factor, when parents (but not friends) are religious, students tend to be more religious than when friends (but not parents) are religious. Although not significant, Politics shows a similar pattern, consistent with the hypothesis that religiosity and politics are transmitted vertically [49].

**Table 8. Interpretation of the two values of each factor after dichotomisation.**

| Factor | Trait | Opposite Trait |
|---|---|---|
| Religiosity | Religious | Non-religious |
| Politics | Right-wing | Left-wing |
| Environment | Environmentalist attitudes and behaviours | No environmentalist attitudes and behaviours |
| Health | Pro-health attitudes and behaviours | No pro-health attitudes and behaviours |
| Music | Listens and enjoys music a lot | Listens and enjoys music little |
| Reading | Practises and enjoys reading a lot | Practises and enjoys reading little |
| Screen | Watches and enjoys screens a lot | Watches and enjoys screens little |
| Social | High social activity | Low social activity |

**Table 9. The calculation of transmission coefficients illustrated with factor Environmentalism.**

| Factor | PT Parental Type | P Nr networks | N Nr students with trait | P−N Nr students with opposite trait | B=N/P Transmission coefficient (trait) | B = (P−N)/P Transmission coefficient (opposite trait) |
|---|---|---|---|---|---|---|
| Environ. | VH | 156 | 111 | 45 | 0.712 | 0.288 |
| Environ. | Vh | 119 | 61 | 58 | 0.513 | 0.487 |
| Environ. | vH | 122 | 66 | 56 | 0.541 | 0.459 |
| Environ. | vh | 185 | 71 | 114 | 0.384 | 0.616 |

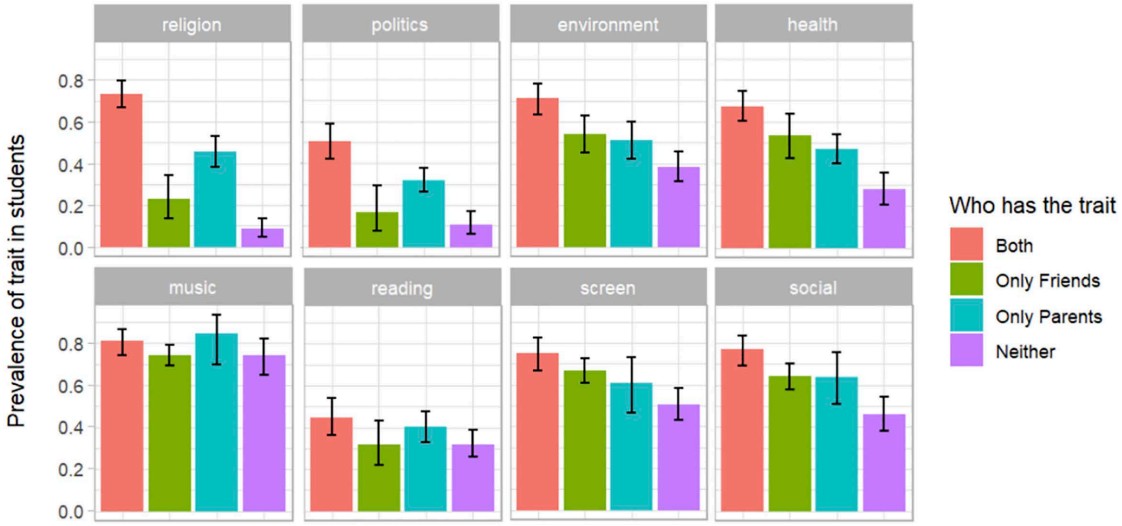

**Fig 5. Proportion of students in each cultural parental type who have the trait for each factor with bootstrapped 95% CIs.**

**Table 10. Vertical, horizontal and other bias values for each trait and opposite trait.**

| Factor | Vert. Bias | Horiz. Bias | Other Bias (Trait) | Other Bias (Opposite Trait) |
|---|---|---|---|---|
| Religiosity | 0.42 | 0.18 | 0.1 | 0.3 |
| Politics | 0.19 | 0.21 | 0.14 | 0.46 |
| Environ. | 0.22 | 0.18 | 0.33 | 0.27 |
| Health | 0.08 | 0.21 | 0.36 | 0.35 |
| Music | 0.07 | 0 | 0.7 | 0.23 |
| Reading | 0.08 | 0.04 | 0.36 | 0.56 |
| Screen | 0.09 | 0.22 | 0.45 | 0.24 |
| Social | 0.07 | 0.15 | 0.51 | 0.27 |

We used maximum likelihood estimation to obtain the strength of our students' biases to resemble their parents (vertical), their friends (horizontal) or neither (other) (Table 10). Vertical and Horizontal biases are the same for the trait and the opposite trait because of the complementarity of the values.

The proportional strengths of the Vertical, Horizontal and Other biases for each trait/opposite trait are plotted in Fig 6 (for alternative visualisation, see Figure A in S9 Text). Although the Vertical and Horizontal biases are the same for a trait and its opposite, the proportional strengths can differ due to differences in Other biases (e.g., the Vertical bias of .19 for Politics is proportionally stronger for the right-wing trait, with Horizontal and Other biases of .21 and .14, than for the left-wing trait, which has Horizontal and Other biases of .21 and .46).

Biases indicating the provenance of information vary substantially across factors. The high proportional strength of Vertical and Horizontal biases for being religious or non-religious, and for having a right-wing political ideology (Table 10 and Fig 6), support the strong influence of those relations for these factors. On the other hand, the low proportional strength of Vertical and Horizontal biases for appreciation of music, reading and social activities, and being left wing, indicate the tendency to acquire those traits from sources other than parents and peers (or at least other than the parents and peers who completed the survey), which may include the media, including social media [59], agents such as other relatives and

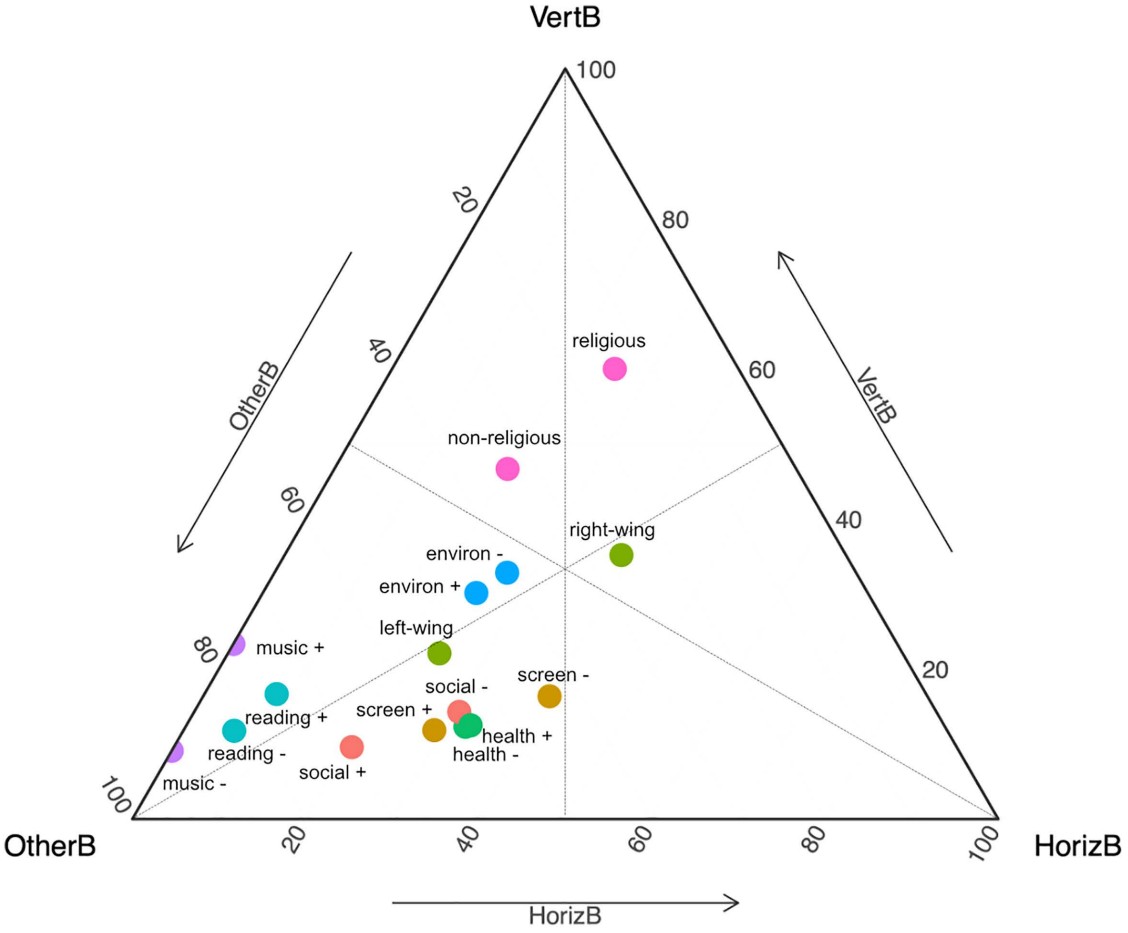

**Fig 6. Vertical, Horizontal and Other bias values for the trait and the opposite trait related to each factor.** Bias strength is proportional to the closeness of the coloured points to the relevant corner of the triangle. E.g., religious students were biased to resemble their parents' religiosity most, then their friends and, finally, others' religiosity ('religious' is closest to the VertB corner, then to the HorizB corner and furthest from OtherB corner); students who did not enjoy music were biased to resemble others' music enjoyment much more than their parents', and not biased to resemble their friends' music enjoyment at all (music- is closest to OtherB, then to VertB, and maximally far from HorizB corner); students enjoying music were biased to resemble their parents' musical enjoyment a little more ('music +' is a little closer to VertB). The bias values defining each point (Table 10) have been normalised to fit in the triangle plot.

friends, teachers and other acquaintances, as well as, potentially, individual learning, in other words, innovation by the students themselves.

## Discussion

This study examined patterns of attitudes and behaviours within social networks using a large data set from students (N = 1905) and their family and friends (N = 4000), each completing a survey that included 27 questions. However, the findings should be interpreted with caution given several limitations. Its correlational design precludes causal conclusions, as observed associations may reflect shared background factors. The reliance on a student convenience sample from a single Australian university, along with gender imbalance and low, uneven recruitment of contacts, also limits generalizability.

We analysed our rich data set from different perspectives, grouping our survey questions into 8 factors (but also by question: S5 Text, S8 Text). Path analysis assumes vertical transmission from grandparents to parents to students,

and directional transmission from among friends and students. Correlational and resemblance analyses do not assume directionality and explore all relations, including horizontal transmission between couples and between the student and their 2 friends, as well as cross-generational (grandparents – student) and at the level of a small social network (around students).

While we acknowledge that our data does not allow us to make causal inferences, our findings are consistent with a range of causal mechanisms, involving cultural transmission, biological inheritance, psychological alignment, and environmental exposure. We explore five key patterns considering these potential mechanisms.

First, a clear conclusion from our data is that not all behaviours and attitudes are transmitted equally; in alignment with [13], strong similarity is largely restricted to religiosity, political ideology (including environmentalism as a component), and health-related behaviours, while most other factors show negligible correlation even among close relations. This pattern aligns with theoretical models like Dual-Inheritance Theory [32] and gene-culture coevolution [34], which suggest that the degree of biological, cultural, and environmental influence differs by domain [33]. For example, we find little evidence of similarity in behaviours such as media consumption or leisure activities, suggesting these may be more context-dependent or idiosyncratic. In contrast, identity-defining religious and political attitudes appear more likely to correlate across generations and social networks.

A second key finding is the strength of vertical similarity between parents and children and, in some cases, also between grandparents and parents, particularly in religiosity, environmentalism, political beliefs, and health behaviours. Both genetic and social-environmental mechanisms could be at play. Parental religiosity is one of the most reliable predictors of religiosity in children across cultural backgrounds [12,54–56]. In this domain, twin studies show very strong genetic influences in adulthood, but not in childhood, suggesting that socialisation plays a larger role early in life [60,61]. A similar pattern is apparent for political orientation. Genetic studies reveal substantial heritable components in adult political attitudes [62,63] which are also linked to genetically-influenced traits such as temperament and personality, observable in early childhood [64,65]. On the other hand, longitudinal studies show that early parenting styles [66,67] and socialisation with parents [68] influence political ideology, particularly in adolescence. In the domain of health behaviours, parent-child similarity is widely documented, e.g., for physical activity [69], diet [70], and healthcare use [71]. High heritability has been found for exercise [72] and diet [73–75]. Again, genetic influences are stronger later in life, while parenting style, family dynamics, and communication quality explain more variation in children and adolescents [76,77]. Vertical similarity values are strongest for the parent-child dyad, but in some cases also appear in the grandparent-parent relationship, suggesting that the multigenerational continuity within families in religiosity, politics, including environmentalism, and health, is underpinned by a strong genetic component, which may be masked during childhood and adolescence by social influences. This may help explain the stronger associations in our data between the (relatively younger) students and their parents than between the parents (relatively older) and their own parents. As an additional cause for similarity, the possibility of reverse vertical transmission, from children to parents or to grandparents, cannot be discounted, especially in areas such as environmentalism, where younger people tend to have higher awareness or expertise. Recent evidence of 'reverse socialisation' in product consumption [78], technology and digital knowledge [79] and, especially, in pro-environmental attitudes and behaviours [80,81], suggests that, e.g., the significant relationship between students and grandparents, and students and parents (Table 7) in environmentalism might be due to this social transmission pathway.

Thirdly, we observe high horizontal similarity between mother and father across all factors, and in each grandparent couple for many factors (Fig 4). These results might be explained by assimilation (convergence in the couple due to mutual influence over time), or by belonging to the same environment, which facilitated their meeting, but existing literature suggests a stronger role for homophily, or assortative mating: the tendency to choose a partner who is already similar to you in beliefs and values. Religious and sociopolitical similarity are among the strongest predictors of mate selection, surpassing even physical or personality traits, and that such similarity arises more from partner selection than

from convergence over time, implying that the attitudes shared by parents (and grandparents) may reflect shared starting points rather than mutual shaping [82]. Our intergenerational data, however, is also consistent with assimilation. We see few significant resemblance links between maternal and paternal grandparents for Religiosity, Politics and Environmentalism (Fig 4), suggesting that assortative mating along those dimensions in the parental couple was not very strong, leaving convergence over the time together as a potential explanation for mother-father similarity. Horizontal similarity in couples is important because it may reinforce intergenerational correlation, providing a consistent environment in which children are socialised. At the same time, this explanation makes it difficult to disentangle which parent exerts more influence on the child, or whether the similarity arises from an aligned household environment.

In fourth place, our data show another horizontal similarity pattern, whereby students tend to resemble their friends, particularly in core domains like religiosity, political beliefs, and environmental concern. Here, again, homophily, or peer selection, is a well-supported psychological mechanism that helps explain why individuals within a peer group often hold similar values and engage in similar behaviours [38]. Additionally, cultural transmission between the friends may explain high convergence on similar ideas after friendships are formed. Mutual influence is key in adolescent friendship networks [37], as documented by religious and ideological clustering within adolescent peer groups [83,84]. Thus, the resemblance we observe among friends may arise from two cultural mechanisms: selection into friendships and cultural transmission within them.

Finally, we found a broader pattern of clustering across the entire network, even among agents who are not directly connected, e.g., between students' friends and students' grandparents. Further, we found similar correlation levels along vertical and horizontal lines for Religiosity and Politics (Table 5). This suggests that cultural traits are not only shared dyadically, but also shaped by broader network structures. Several mechanisms may explain this. One possibility is compound assortativity, where people choose friends who resemble their families, either directly or indirectly. Another is environmental structuring, where shared neighbourhoods, schools, or communities expose individuals to similar information and norms. Modern friendship and mate choice are both governed by sociopolitical similarity, which creates dense networks of aligned religious beliefs and political attitudes, even in the absence of direct contact [82]. These mechanisms may interact to produce cultural coherence at the network level, consistent with our findings.

The patterns of behavioural similarity in our data are best explained by a combination of mechanisms (Table 11). In some cases, these mechanisms are clearly supported by prior research (e.g., vertical transmission of religiosity and politics); in others, they offer plausible accounts grounded in known psychological and sociological processes (e.g., assortative mating). Our results do not point to any single cause but, rather, suggest that behavioural resemblance reflects the dynamic interplay of biological and cultural inheritance.

Extended data collection across different countries and cultures would help determine how universal these patterns are and explore effects and interactions of, e.g., prevalent religion or political system. Larger, more gender-balanced datasets would allow measuring gender effects such as maternal and paternal effects or gender congruency. Longitudinal data collection and experiments would help test the causal roles of homophily, assimilation and transmission on similarity, while

**Table 11. Main resemblance patterns within members of the networks observed in our data and main suggested explanations.**

| Factor | Main resemblance pattern | Main explanation |
|---|---|---|
| Religiosity, Politics & Environmentalism | Strong resemblance across generations and within social networks | Early in life: socialisation<br>Later in life: strong genetic heritability<br>Assortative mating/ peer selection, environmental structuring |
| Health | Resemblance across generations and within social networks | Early in life: parenting style, family dynamics<br>Later in life: Strong genetic heritability |
| Music, Reading, Screen, & Social | Little resemblance observed | Little family and friend influence |

genetic studies will continue to demarcate the contribution of genes, upbringing and other environmental factors over the lifespan. Long-term social media analysis could, for instance, reveal examples of deep vertical transmission over many generations within families and other groups, and of horizontally transmitted traits that die off with their cohort. Exploring the fast-changing landscape of social media information platforms may reveal not only current changes but cyclical or other long-term patterns of change that could be extrapolated to the future. Quantitative data can be profitably complemented with qualitative studies which may suggest further causes of similarity and help answer some of the question posed by our data, e.g., why do non-religious and left-wing student align with sources *other* than their parents and friends' religiosity and political views more than religious and right-wing students?

Understanding the mechanisms behind similarity, contagion and cultural transmission across cultural domains should improve the efficacy of interventions aimed at changing specific attitudes and behaviours. For example, school and university campaigns and activities may be designed to educate older generations on fast-changing topics including environmentalism and new technologies. Ultimately, predicting and controlling the spread of behaviour and attitudes will require integrating data about behaviour and attitude similarity and social network structure within a theoretical framework such as dual inheritance theory, comprising social, psychological and biological transmission mechanisms.

## Supporting information

**S1 Text. Network size distribution.**
(PDF)

**S2 Text. The survey used in this study.**
(PDF)

**S3 Text. Cavalli-Sforza et al.'s (1982) survey.**
(PDF)

**S4 Text. Numeric recoding answers to text questions.**
(PDF)

**S5 Text. Descriptive statistics of numeric answers.**
(PDF)

**S6 Table. Factor loadings.**
(PDF)

**S7 Text. Descriptives of text-based answers.**
(PDF)

**S8 Text. Resemblance by question.**
(PDF)

**S9 Text. Parental type analysis and maximum likelihood estimation of vertical, horizontal and other biases.**
(PDF)

**S10 Text. Prevalence of traits in students by parental type by question.**
(PDF)

## Author contributions

**Conceptualization:** Monica Tamariz, Nicolas Fay.

**Data curation:** Monica Tamariz, Bradley Walker, Matthew Bennett.

**Formal analysis:** Monica Tamariz, Bradley Walker, José Segovia-Martín.

**Investigation:** Bradley Walker, Matthew Bennett.

**Methodology:** Monica Tamariz, Bradley Walker, José Segovia-Martín, Nicolas Fay.

**Project administration:** Monica Tamariz, Nicolas Fay.

**Software:** Monica Tamariz, Bradley Walker, José Segovia-Martín.

**Supervision:** Nicolas Fay.

**Validation:** Monica Tamariz, Bradley Walker.

**Visualization:** Monica Tamariz, Bradley Walker.

**Writing – original draft:** Monica Tamariz, Matthew Bennett.

**Writing – review & editing:** Monica Tamariz, Bradley Walker, José Segovia-Martín, Nicolas Fay.

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
