## [Decision Letter · Decision Letter 0]

17 Sep 2025

Dear Dr. Tamariz

Thank you for submitting your manuscript to PLOS ONE. I recieved two referee reports recommending revisions. After careful consideration, we feel that it has merit but does not fully meet PLOS ONE’s publication criteria as it currently stands. Therefore, we invite you to submit a revised version of the manuscript that addresses the points raised during the review process.

We look forward to receiving your revised manuscript.

Kind regards,

Francesco Flaviano Russo

Academic Editor

PLOS ONE

Journal Requirements:

Additional Editor Comments:

Dear Dr. Tamariz

I have received two referee reports recommending a revision of the paper. I invite you to revise the paper along the lines suggested by both of them, and to resubmit the paper.

Best,

Francesco Flaviano Russo

Reviewers' comments:

Reviewer's Responses to Questions

**Comments to the Author**

1. Is the manuscript technically sound, and do the data support the conclusions?

Reviewer #1: Yes

Reviewer #2: Yes

2. Has the statistical analysis been performed appropriately and rigorously?

Reviewer #1: Yes

Reviewer #2: Yes

3. Have the authors made all data underlying the findings in their manuscript fully available?

Reviewer #1: Yes

Reviewer #2: Yes

4. Is the manuscript presented in an intelligible fashion and written in standard English?

Reviewer #1: Yes

Reviewer #2: Yes

Reviewer #1: Referee Report

Manuscript Number: PONE-D-25-34652

Title: Cultural transmission of attitudes and behaviours from parents, peers and grandparents

Journal: PLOS ONE

Overall Recommendation: Major Revision

General Comments:

Thank you for the opportunity to review this manuscript. This study presents a large-scale, multi-method conceptual replication and extension of the seminal work by Cavalli-Sforza et al. (1982). The ambition of the project is commendable, and the dataset of nearly 6000 individuals is a significant strength that provides considerable power. The application of modern analytical techniques (path analysis, Monte Carlo simulations) to the classic question of cultural transmission pathways is a valuable contribution to the field.

The manuscript is generally well-structured, the analytical methods are sophisticated and mostly well-described, and the results are presented clearly. The discussion thoroughly engages with the complex, multifactorial nature of cultural resemblance.

However, I have significant concerns regarding the generalizability of the findings due to the sampling strategy and what I perceive as an occasional overinterpretation of the correlational data. The claims of novelty, particularly regarding replication of Cavalli Sforza may need to be tempered. Addressing these concerns would significantly strengthen the manuscript and its contribution to the literature.

Major Comments:

1. Sampling and Generalizability:

The use of a convenience sample of university students from a single Australian institution introduces substantial selection bias (age, socioeconomic status, education level, cultural background). While this method is pragmatic for recruiting social networks, it severely limits the generalizability of the findings to the broader population. The pronounced gender imbalance (72.2% women in the student sample) further compounds this issue. The manuscript should more explicitly and prominently acknowledge this limitation throughout, particularly in the abstract and discussion, and caution against claims of describing "contemporary populations" without qualification.

2. Claims of Novelty and Replication:

The assertion that this is a replication study, potentially framed as the "first," is inaccurate. The work of Cavalli-Sforza and Feldman (1982) has spawned an entire field of cultural evolution, with countless studies testing, applying, and refining its models (e.g., the work of Boyd, Richerson, Henrich, and others). The literature review, while good, does not adequately frame this study within that extensive existing body of work. The manuscript's valuable contribution is not as the first replication but as a large-scale, modern replication with an extended social network (grandparents) and advanced methods. The language should be revised to reflect this more accurate positioning.

3. Interpretation and Causal Language:

The discussion is comprehensive but at times risks overinterpreting the correlational data. Although the authors correctly note on page 39 that the study does not test causal mechanisms, the subsequent discussion frequently uses causal language ("transmission," "influence," "exert influence") when discussing results. While these terms are standard in the field, the manuscript would benefit from a more consistent and cautious phrasing that emphasizes the interpretation of patterns of resemblance for which transmission is one of several plausible mechanisms. The discussion of mechanisms (genetics, homophily, transmission) is excellent but should be framed more consistently as speculative interpretation rather than established conclusion.

4. Response Rate and Attrition Bias:

The manuscript does not explicitly state the response rate for the recruitment of family and friends by the student participants. Furthermore, the exclusion of 722 student networks because they contained no family or friends potentially introduces a significant attrition bias. These excluded students may systematically differ from those in the analyzed networks (e.g., weaker family ties, international students). This potential bias should be acknowledged as a limitation.

Minor Comments:

Methods (Factor Analysis): Please consider reporting common fit indices (e.g., RMSEA, CFI) for the factor analysis solution to provide a standard measure of how well the 8-factor model fits the data.

Methods (Additive Model): The dichotomization of continuous factor scores, while faithful to the original 1982 method, is a loss of information. A brief sentence acknowledging this as a limitation of this particular analysis would be appropriate.

Discussion (Readability): The discussion is highly informative but very dense. I strongly suggest adding a summary table that lists the key findings for each major factor (e.g., Religiosity, Politics) and the authors' leading hypothesized explanation for the observed resemblance (e.g., "Vertical transmission + genetic heritability"; "Spouse similarity: Assortative mating"). This would greatly enhance clarity and readability.

Data Availability: The data is stated to be on OSF, but the provided link (https://osf.io/v8hc7) in the text points to a project that is not found (404 error). This must be corrected before publication.

Suggested Revisions:

Abstract & Introduction: Temper claims of novelty and replication. Rephrase to highlight the study as a large-scale, modern extension.

Methods (Section 2.1): Add a paragraph explicitly discussing the limitations of the student convenience sample and the potential attrition bias from excluding disconnected networks.

Results: Ensure all links in the data availability statement are functional.

Discussion:

Begin the discussion with a paragraph that explicitly lists the main limitations: the correlational design, the sampling strategy, and the attendant restrictions on generalizability and causal inference.

Consistently use cautious language that frames explanations as hypotheses (e.g., "one possible explanation is," "our results are consistent with").

Add a summary table as suggested to organize the key interpretations.

Conclusion: Scale back broad generalizations. Clearly state that the findings are most applicable to similar demographic cohorts (Western, educated, young adult populations).

Final Judgment:

The study presents a valuable dataset and a robust set of analyses. However, the issues of sampling, generalizability, and framing are substantial and must be addressed before the manuscript can be considered for publication. I therefore recommend Major Revision. I am enthusiastic about the potential of this work once these concerns are adequately mitigated and believe it will be a strong contribution to PLOS ONE after revisions.

Reviewer #2: 1. The study presents a clear problem that advances the seminal work of Cavalli-Sforza et al in current environment and under much multifactorial perspective to establish the vertical,horizontal and oblique relationships to help understand the cultural transmission of attitudes and behaviours from parents, peers and grandparents. The research gap established are well situated and justified. The undergirding research questions are legitimate and rooted in well grounded theoretical framing in cultural studies central to cultural evolutionary research.

2. The methodological plan and analysis (descriptive statistics, exploratory factor analysis, path analysis and simulation) have been well laid out and every phase including the simulations are academically-rigorous and would enable replication of the study in similar contexts.

3. The discussions of the results are interesting and sound. There are no invalidated claims. There are rich and academically-refreshing comparisons and contradictions of the results in the current study with those in the extant literature which are highly remarkable of the study. However, the discussion would have been enriched if intentional comparison of the study's results in relation to different country contexts are presented rather than a broad or general approach adopted by the authors.

4. I don't know the reason for the silence of the paper on the geographical scope (study area) of the study. I think this would give important contextual understanding of the results.

5. The suggestions for further research are great and offers enough directional paths. I was wondering if the results are not too pre-mature as I belive they are, some realistic recommendations are drawn from the study's results for policy and practice.

**Do you want your identity to be public for this peer review?** For information about this choice, including consent withdrawal, please see our Privacy Policy

Reviewer #1: **Yes:** Marcello D'Amato

Reviewer #2: **Yes:** Dickson Adom

---

## [Author Response · Author response to Decision Letter 1]

23 Sep 2025

(It will be easier to read these responses in the uploaded response to reviewers letter.)

Dear Editor,

Thank you for the opportunity to revise our manuscript. Please find below our detailed responses to the reviewers’ comments. We have addressed each point carefully, indicated the changes made to the manuscript and provided clarifications to explain our decisions.

We appreciate the reviewers’ constructive feedback, which has helped us improve the clarity and rigor of the manuscript, as well as the cross-cultural outlook.

Sincerely,

Monica Tamariz

Comment Response

Editor

I have received two referee reports recommending a revision of the paper. I invite you to revise the paper along the lines suggested by both of them, and to resubmit the paper.

No response needed

Reviewer #1

Thank you for the opportunity to review this manuscript. This study presents a large-scale, multi-method conceptual replication and extension of the seminal work by Cavalli-Sforza et al. (1982). The ambition of the project is commendable, and the dataset of nearly 6000 individuals is a significant strength that provides considerable power. The application of modern analytical techniques (path analysis, Monte Carlo simulations) to the classic question of cultural transmission pathways is a valuable contribution to the field.

The manuscript is generally well-structured, the analytical methods are sophisticated and mostly well-described, and the results are presented clearly. The discussion thoroughly engages with the complex, multifactorial nature of cultural resemblance.

However, I have significant concerns regarding the generalizability of the findings due to the sampling strategy and what I perceive as an occasional overinterpretation of the correlational data. The claims of novelty, particularly regarding replication of Cavalli Sforza may need to be tempered. Addressing these concerns would significantly strengthen the manuscript and its contribution to the literature. We thank Prof. D’Amato for his careful assessment of our manuscript, positive general comments and praise for the sample size and analyses. We acknowledge the concerns he points out and explain below how we address them.

Major comment 1. Sampling and Generalizability:

The use of a convenience sample of university students from a single Australian institution introduces substantial selection bias (age, socioeconomic status, education level, cultural background). While this method is pragmatic for recruiting social networks, it severely limits the generalizability of the findings to the broader population. The pronounced gender imbalance (72.2% women in the student sample) further compounds this issue. The manuscript should more explicitly and prominently acknowledge this limitation throughout, particularly in the abstract and discussion, and caution against claims of describing "contemporary populations" without qualification.

Suggested revision 2. Methods (Section 2.1): Add a paragraph explicitly discussing the limitations of the student convenience sample and the potential attrition bias from excluding disconnected networks.

Suggested revision 7. Scale back broad generalizations. Clearly state that the findings are most applicable to similar demographic cohorts (Western, educated, young adult populations). We now acknowledge these limitations:

In the Abstract, “in contemporary populations” has been replaced with “in a contemporary population”.

We have added this paragraph in section 2.1, L 233-260: “Our reliance on a student convenience sample limits the generalizability of the findings, as students may differ from the broader population in demographics, network structure, and social experiences. Data from a single Australian university also reflects limited geographical and cultural diversity, which may constrain the applicability of the results to other settings. In addition, there was a notable gender imbalance, with women being overrepresented, which may influence the observed patterns of social transmission. Recruitment of family and friends by student participants was relatively low, with many students not recruiting anyone and, among those who did, most recruiting only a small number of contacts. Finally, excluding networks where no family or friends participated may have introduced attrition bias, potentially underrepresenting socially isolated individuals and networks with different transmission dynamics."

Major comment 2. Claims of Novelty and Replication:

The assertion that this is a replication study, potentially framed as the "first," is inaccurate. The work of Cavalli-Sforza and Feldman (1982) has spawned an entire field of cultural evolution, with countless studies testing, applying, and refining its models (e.g., the work of Boyd, Richerson, Henrich, and others). The literature review, while good, does not adequately frame this study within that extensive existing body of work. The manuscript's valuable contribution is not as the first replication but as a large-scale, modern replication with an extended social network (grandparents) and advanced methods. The language should be revised to reflect this more accurate positioning.

Suggested revision 1. Abstract & Introduction: Temper claims of novelty and replication. Rephrase to highlight the study as a large-scale, modern extension. We have downplayed the replication aspect of our work and the novelty of our questions and data in section “The present study” by greatly simplifying the first paragraph of this section and removing all mentions of replication (L 186-187).

We have also specified how our study “builds on” Cavalli-Sforza & Feldman’s work (L 193), and removed an allusion to their work’s limitations (L 193).

At the end of the same section, we replaced “novel data” with the weaker “a new, modern data set” (L224).

Major comment 3. Interpretation and Causal Language:

The discussion is comprehensive but at times risks overinterpreting the correlational data. Although the authors correctly note on page 39 that the study does not test causal mechanisms, the subsequent discussion frequently uses causal language ("transmission," "influence," "exert influence") when discussing results. While these terms are standard in the field, the manuscript would benefit from a more consistent and cautious phrasing that emphasizes the interpretation of patterns of resemblance for which transmission is one of several plausible mechanisms. The discussion of mechanisms (genetics, homophily, transmission) is excellent but should be framed more consistently as speculative interpretation rather than established conclusion.

Suggested revision 5. Discussion: Consistently use cautious language that frames explanations as hypotheses (e.g., "one possible explanation is," "our results are consistent with"). We have removed remaining causal terms from the discussion of our results (but have left them in when they refer to reviewed literature that shows causal links), and make our language more cautious:

Replacements:

L 735 persist  correlate

L 740 mechanisms seem to be at play  mechanisms could be at play

L772 these results could be explained  might be explained

L 786 transmission  correlation

L 797 cultural transmission between the friends may increase convergence  cultural transmission between the friends may explain high convergence

L 806 transmitted  shared

Major comment 4. Response Rate and Attrition Bias:

The manuscript does not explicitly state the response rate for the recruitment of family and friends by the student participants. Furthermore, the exclusion of 722 student networks because they contained no family or friends potentially introduces a significant attrition bias. These excluded students may systematically differ from those in the analyzed networks (e.g., weaker family ties, international students). This potential bias should be acknowledged as a limitation. Response rate for recruitment of family and friends is now given as a table in Supplementary Information S1. We have added this to Participants section (L 256): “Recruitment of family and friends by student participants was relatively low, with many students not recruiting anyone and, among those who did, most recruiting only a small number of contacts.”

Attrition bias is addressed in this addition to the same section (L 258): “Finally, excluding networks where no family or friends participated may have introduced attrition bias, potentially underrepresenting socially isolated individuals and networks with different transmission dynamics.”

Minor comment 1. Methods (Factor Analysis): Please consider reporting common fit indices (e.g., RMSEA, CFI) for the factor analysis solution to provide a standard measure of how well the 8-factor model fits the data. In section Results of factor analysis (L 438-441), we have added: “The factor analysis yielded an eight-factor solution that demonstrated an acceptable overall fit to the data, with good values for the CFI (.95) and RMSEA (.04), and a TLI (.89) that was slightly below the conventional cutoff of .90.”

Minor comment 2. Methods (Additive Model): The dichotomization of continuous factor scores, while faithful to the original 1982 method, is a loss of information. A brief sentence acknowledging this as a limitation of this particular analysis would be appropriate. We added (L 376-78): A limitation of this method is the fact that the dichotomization of continuous factor scores, while faithful to the original method (49), involves a loss of information.

Minor comment 3. Discussion (Readability): The discussion is highly informative but very dense. I strongly suggest adding a summary table that lists the key findings for each major factor (e.g., Religiosity, Politics) and the authors' leading hypothesized explanation for the observed resemblance (e.g., "Vertical transmission + genetic heritability"; "Spouse similarity: Assortative mating"). This would greatly enhance clarity and readability.

Suggested revision 6. Discussion: Add a summary table as suggested to organize the key interpretations. We have included table 11, in line 825:

Table 11. Main resemblance patterns within members of the networks observed in our data and main suggested explanations.

Factor Main resemblance pattern Main explanation

Religiosity, Politics & Environmentalism Strong resemblance across generations and within social networks Early in life: socialisation

Later in life: strong genetic heritability

Assortative mating / peer selection,

environmental structuring

Health Resemblance across generations and within social networks Early in life: parenting style, family dynamics

Later in life: Strong genetic heritability

Music, Reading, Screen, & Social Little resemblance observed Little family and friend influence

Minor comment 4. Data Availability: The data is stated to be on OSF, but the provided link (https://osf.io/v8hc7) in the text points to a project that is not found (404 error). This must be corrected before publication.

Suggested revision 3. Results: Ensure all links in the data availability statement are functional. The data set is publicly available on OSF. We have checked from different browsers and accounts, and the link seems to work fine for us.

Suggested revision 1. Abstract & Introduction: Temper claims of novelty and replication. Rephrase to highlight the study as a large-scale, modern extension. See response to “Major comment 2. “Claims of Novelty and Replication” above.

Suggested revision 2. Methods (Section 2.1): Add a paragraph explicitly discussing the limitations of the student convenience sample and the potential attrition bias from excluding disconnected networks. See response to “Major comment 1. “Sampling and Generalizability” above.

Suggested revision 4. Discussion: Begin the discussion with a paragraph that explicitly lists the main limitations: the correlational design, the sampling strategy, and the attendant restrictions on generalizability and causal inference. We have replaced the first two paragraphs in the discussion with this paragraph (L689-695):

“This study examined patterns of attitudes and behaviours within social networks using a large data set from students (N=1905) and their family and friends (N=4000), each completing a survey that included 27 questions. However, the findings should be interpreted with caution given several limitations. Its correlational design precludes causal conclusions, as observed associations may reflect shared background factors. The reliance on a student convenience sample from a single Australian university, along with gender imbalance and low, uneven recruitment of contacts, also limits generalizability.”

Suggested revision 5. Discussion: Consistently use cautious language that frames explanations as hypotheses (e.g., "one possible explanation is," "our results are consistent with"). See response to “Major comment 1. “Sampling and Generalizability” above

Suggested revision 6. Discussion: Add a summary table as suggested to organize the key interpretations. See response to “Minor comment 3. Discussion (Readability)” above.

Suggested revision 7. Conclusion: Scale back broad generalizations. Clearly state that the findings are most applicable to similar demographic cohorts (Western, educated, young adult populations). See response to “Major comment 1. Sampling and Generalizability” above.

Reviewer #2

1. The study presents a clear problem that advances the seminal work of Cavalli-Sforza et al in current environment and under much multifactorial perspective to establish the vertical,horizontal and oblique relationships to help understand the cultural transmission of attitudes and behaviours from parents, peers and grandparents. The research gap established are well situated and justified. The undergirding research questions are legitimate and rooted in well grounded theoretical framing in cultural studies central to cultural evolutionary research.

2. The methodological plan and analysis (descriptive statistics, exploratory factor analysis, path analysis and simulation) have been well laid out and every phase including the simulations are academically-rigorous and would enable replication of the study in similar contexts.

3. The discussions of the results are interesting and sound. There are no invalidated claims. There are rich and academically-refreshing comparisons and contradictions of the results in the current study with those in the extant literature which are highly remarkable of the study. We thank Dr Adom for his careful assessment of our manuscript and his positive comments. We explain below how we address each of his concerns.

However, the discussion would have been enriched if intentional comparison of the study's results in relation to different country contexts are presented rather than a broad or general approach adopted by the authors.

We have added new references to studies conducted outside western countries for religiosity and political orientation in the discussion, in L 741-751:

75. Lee et al. Parental effects on diet and obesity - 23 countries (high diversity)

55. Milos & Glavas - Parental effects on Religiosity - Croatia

56. Chamratithirong et al. - Parental effects on Religiosity - Thailand

67. Sahertian et al - Parental effects on Religiosity - Indonesia

68. Hong & Lim - Parental socialisation and Political participation - Singapore

4. I don't know the reason for the silence of the paper on the geographical scope (study area) of the study. I think this would give important contextual understanding of the results. The geographical setting is now clearly mentioned in the Participants section:

L 231: “from the University of Western Australia (UWA), in Perth, Australia”

We acknowledge the limitation inherent to our study related to the homogeneity of our sample. We spell this out by emphasising in several places the fact that our data comes from a single university Australia:

L 250 (Methods): “Our reliance on a student convenience sample limits the generalizability of the findings, as students may differ from the broader population in demographics, network structure, and social experiences. Data from a single Australian university also reflects limited geographical and cultural diversity, which may con

---

## [Decision Letter · Decision Letter 1]

7 Jan 2026

Cultural transmission of attitudes and behaviours from parents, peers and grandparents

PONE-D-25-34652R1

Dear Dr. Tamariz,

We’re pleased to inform you that your manuscript has been judged scientifically suitable for publication and will be formally accepted for publication once it meets all outstanding technical requirements.

Kind regards,

Francesco Flaviano Russo

Academic Editor

PLOS One

Reviewer #2: All comments have been addressed

2. Is the manuscript technically sound, and do the data support the conclusions?

Reviewer #2: Yes

3. Has the statistical analysis been performed appropriately and rigorously?

Reviewer #2: Yes

4. Have the authors made all data underlying the findings in their manuscript fully available?

Reviewer #2: Yes

5. Is the manuscript presented in an intelligible fashion and written in standard English?

Reviewer #2: Yes

Reviewer #2: Thank you for satisfactorily addressing the issues I raised and for clarification on the recommendation section. The added references have indeed expanded the dialogue to other regions which would enhance the study's relevance in the scholarship. Thanks and best regards

**Do you want your identity to be public for this peer review?** For information about this choice, including consent withdrawal, please see our Privacy Policy

Reviewer #2: **Yes:** Dickson Adom

---

## [Editor Report · Acceptance letter]

PONE-D-25-34652R1

PLOS One

Dear Dr. Tamariz,

I'm pleased to inform you that your manuscript has been deemed suitable for publication in PLOS One. Congratulations! Your manuscript is now being handed over to our production team.

Kind regards,

on behalf of

Professor Francesco Flaviano Russo

Academic Editor

PLOS One